# Artificial Patination of Copper and Copper Alloys in Wet Atmosphere with Increased Content of SO₂

**Richard Bureš [1], Martin Klajmon [2], Jaroslav Fojt [1], Pavol Rak [1], Kristýna Jílková [3] and Jan Stoulil [1,\*]**

[1] Department of Metals and Corrosion Engineering, University of Chemistry and Technology, 166 28 Prague, Czech Republic; buresr@vscht.cz (R.B.); fojtj@vscht.cz (J.F.); rakp@vscht.cz (P.R.)

[2] Department of Physical Chemistry, University of Chemistry and Technology, 166 28 Prague, Czech Republic; klajmonm@vscht.cz

[3] Department of Glass and Ceramics, University of Chemistry and Technology, 166 28 Prague, Czech Republic; rysovak@vscht.cz

\* Correspondence: stoulilj@vscht.cz; Tel.: +420-220-443-750

**Abstract:** Natural copper patina is usually formed over several decades. This work investigates the possibility of obtaining a stable artificial patina based on brochantite in a more reasonable time. The patination process was based on patina formation from a humid atmosphere containing sulphur dioxide. The studied parameters were humidity (condensation and condensation/drying), sulphur dioxide concentration (4.4–44.3 g·m$^{-3}$) and surface pre-treatments (grinding, pre-oxidation and pre-patination) prior to the patination process. Samples were evaluated by mass change, digital image analysis, spectrophotometry, scanning electron microscopy (SEM), X-ray photoelectron spectroscopy (XPS) and X-ray diffraction (XRD). A resistometric method was employed in order to observe the patina formation continuously during the exposure. Conditions inside the chamber were monitored during the exposure (pH of water and concentration of SO₂ in gaseous phase). According to XRD, it was possible to deliberately grow a brochantite patina of reasonable thickness (approx. 30 μm), even within a couple of days of exposure. The drying phase of the condensation cycle increased the homogeneity of the deposited patina. Formation kinetics were the fastest under a condensation/drying cycle, starting with 17.7 g·m$^{-3}$ sulphur dioxide and decreasing dosing in the cycle, with an electrolyte pH close to 3. The higher sulphur dioxide content above 17.7 g·m$^{-3}$ forms too aggressive a surface electrolyte, which led to the dissolution of the brochantite. The pre-oxidation of copper surface resulted in a significant improvement of patina homogeneity on the surface.

**Keywords:** copper patina; gaseous phase; artificial patina

## 1. Introduction

The typical colour of clean copper is usually salmon-pink. After the exposure of copper to the atmosphere, this colour starts to change. These changes of colour are due to the formation of a protective layer called patina, which lends an aesthetic and historical value to the object. A formation of patina usually takes several decades under common outdoor conditions. Patina forms due to an interaction of copper with gaseous, aerosolised or solid particles of corrosive species, such as SO₂, NO₂, O₃ and Cl⁻. Changes start with the darkening of the copper surface to brown or even black locally and finish with the formation of an aesthetically optimal green layer. The resulting products (patina) are insoluble. Patina is a layer consisting of corrosion products formed and retained on the copper surface. The rate of formation, composition and colour of patina is influenced by composition of the atmosphere, the level of air pollution and an alternation of wet and dry cycles [1–7].

Currently, the rate of air pollution has decreased compared to previous decades, which correlates with an increase of time to patination [8,9]. Natural patina is spatially heterogeneous and has different layers of the copper corrosion products. This was proven during the study of over 300-year-old roof tiles from Queen Anne's Summer Palace in Prague and of the surface of a bronze monument to Francis Garnier, Paris [10,11]. In the beginning of an exposure, the layer of cuprous oxide ($Cu_2O$), called cuprite, appears on a surface of the copper object [12]. A thin layer forms, even under dry conditions, by direct oxidation [13]. This layer changes the colour of copper to brown and dark brown. The thickness of this layer is 5–10 µm [1,8,12–14]. The mechanism of the formation of cuprite is according to Equations (1) and (2).

$$Cu \rightarrow Cu^+ + e^- \tag{1}$$

$$2Cu^+ + H_2O \rightarrow Cu_2O + 2H^+ \tag{2}$$

In the next step the natural patination continues with oxidative dissolution leading to the formation of cupric ions according to Equation (3). The ions react with pollutants in atmosphere and form the green phase of patina. The composition of these phases depends on the pollutants in the atmosphere. In industrial atmosphere, the green copper patina is usually based on sulphates, whereas patina formed close to the ocean is based on chlorides [15].

$$Cu_2O + 2H^+ \rightarrow 2Cu^{2+} + H_2O + 2e^- \tag{3}$$

The sulphates of copper are brochantite ($Cu_4SO_4(OH)_6$), antlerite ($Cu_3(SO_4)(OH)_4$), posnjakite ($Cu_4(SO_4)(OH)_6 \cdot H_2O$), stranbergite ($Cu_5(SO_4)_2(OH)_6 \cdot 4H_2O$), and langite ($Cu_4(SO_4)(OH)_6 \cdot 2H_2O$). Posnjakite, strangbergite and langite are precursors and they can transform to the more stable phases brochantite and antlerite. Copper ions with chlorides can form nantokite (CuCl) instead of cuprite and subsequently form atacamite ($Cu_2Cl(OH)_3$) [12–14,16–18]. The main components of outdoor patinas are cuprite, brochantite, antlerite, posnjakite and atacamite [4,13,18].

Fonseca et al. have studied copper corrosion in urban and marine atmospheric conditions. They observed that patina formed in an urban atmosphere is less porous and more adherent compared to marine patina. Marine patina is heterogeneous resulting in more probable film spalling, probably due to strong winds and the high degree of wetness during almost all of the year [4]. The time to form patina from sulphates takes almost a year but chloride patinas can be found even after a month of exposure [19].

According to Greadel et al. [16] sulphates are more stable than chlorides, so even in a marine atmosphere, we can find more brochantite than atacamite. For example, the patina on the Statue of Liberty, which is surrounded by the sea, has as the most common compounds brochantite and antlerite, which are typically formed in acidic conditions and sheltered areas. Antlerite is even more stable than brochantite at lower pH levels. This composition is probably a consequence of acidic precipitations in New York Harbour [8,19–22].

This work is aimed at forming an artificial patina based on brochantite, as it is the most stable phase of copper patina. Brochantite is formed on the top of the cuprite layer, and the thickness is usually 5–40 µm. Brochantite is formed from cuprite according to Equation (4) [1].

$$2Cu_2O + SO_4^{2-} + 4H_2O \rightarrow CuSO_4 \cdot 3Cu(OH)_2 + 2H^+ + 4e^- \tag{4}$$

Brochantite is the main phase of copper patina in atmospheres polluted by $SO_2$ and it seems that the amount of brochantite depends on the concentration of $SO_2$ in the atmosphere. However, there is a limit concentration of $SO_2$ above which brochantite does not form and too aggressive a layer of aqueous electrolyte dissolves the patina on the copper surface. In the work of Zittlau [23], there is a range where brochantite is thermodynamically stable. Thus, acid rains can be harmful for patina instead of supporting the formation of a new aesthetic layer on copper. For the artificial ageing of copper by artificial acid, the rain is commonly a solution with pH < 4.5 [18,21,24]. According to the

work of Nassau, the limit of pH for the dissolution of copper patina and other copper compounds is 2.4 [18].

The important role of $SO_2$ in the atmosphere is due to homogeneous and heterogeneous oxidation of sulphur dioxide to sulphuric acid. The homogeneous oxidation primarily starts through the reaction with a hydroxyl radical. The heterogeneous oxidation mainly occurs in cloud droplets or at air–water interfaces of aerosols [25]. The solubility of $SO_2$ is determined by the following dissociation processes in droplets as shown in Equations (5)–(7) [26].

$$SO_2 \text{ (g)} \rightarrow SO_2 \text{ (aq)} \tag{5}$$

$$H_2O + SO_2 \text{ (aq)} \leftrightarrow HSO_3^- + H^+ \tag{6}$$

$$HSO_3^- \leftrightarrow SO_3^{2-} + H^+ \tag{7}$$

Currently, many ways of the artificial patination exist. It is used in architecture, restoration or experimentally. Artificial patina is mostly based on chlorides and nitrates. Basic copper nitrate (gerhardite, $Cu_2NO_3(OH)_3$) has low stability and it transforms readily to other corrosion products [27]. Chlorides are very stable, but their colour is somewhat bluish compared to brochantite. Common patination processes combine treatment by heating, immersion in solution, exposure in acidic water vapour or the application of acidic pastes [3,5,28].

There are also minor procedures for sulphates, but they usually show low reproducibility. This work is focused on patina formation in a wet atmosphere with increased $SO_2$ content, due to lack of knowledge of stable artificial patina based on sulphates, especially brochantite, and different conditions were tested.

Compared to the other methods, patination from gaseous phase represents a process of patina formation more similar to the natural one, which should theoretically render patina with better properties, such as color, adhesion and durability. These properties are given by the composition of patina and the artificial patina formed in condition of atmosphere with increased $SO_2$ content would be based on sulphates like the natural patina formed usually in industrial areas. Sulphate patina is very stable in atmospheric conditions and the main phase is brochantite [1,8,16].

## 2. Materials and Methods

This work investigates formation of copper patina based on brochantite from a gaseous phase. According to the studies of atmospheric corrosion of copper, specific conditions such as level of pollution, level of precipitation and time of exposure are required [4,29]. In order to reach right conditions for the experiment, a special chamber was designed at our department. The corrosion chamber has a volume of 30 $dm^3$ and it is made of polypropylene. It allowes the control of temperature by rod heater with thermostat, blowing air by ventilator placed inside the chamber, humidity and the volume of gas ($SO_2$) dosed in the beginning of exposure. $SO_2$ is dosed manually and the volume of gas is set by pipette and manometer with liquid medium. Demineralised water was used as a feed water on the chamber bottom for condensation conditions (100% RH), and the feed water was changed for every new inlet dose of $SO_2$. The volume of the water/solution was 5 $dm^3$ and the level was always above the rod heater, which was placed at the bottom of the chamber. Copper sheets of dimensions $5 \times 5$ $cm^2$ were exposed under the 60° inclination and the effects of the concentration of $SO_2$, pH, cyclic condition and pre-treatments of the copper surface on the kinetics and formation of patina were observed. At least four copper samples were tested under each condition.

Four surface states of samples were studied, ground and three pre-exposed. Samples were ground by P 120 emery paper. Since natural patina has a layered structure, the pre-exposed surface was also employed. Pre-oxidation and pre-patination solutions were chosen due to the possible formation of copper oxides and copper sulphides. Pre-patination was based on immersion into solution 1 (100 mL $NH_3$ (conc.), 33 g $Na_2S \cdot 9H_2O$ and 250 mL demineralised water) or solution 2 (commercial solution

Sulka-K©–based on calcium and potassium polysulphides). Solutions were chosen according to restoration practice as solutions contain sulphur (not chlorides or carbonates) commonly used for patination of copper object. The third type of pre-exposed surface was pre-oxidation in the oven at 150, 200 or 300 °C for up to 6 h.

The first experiment started with 24 h exposure at 40 °C with different inlet doses of $SO_2$ (8.9, 17.7, 26.6, 35.4 and 44.3 $g \cdot m^{-3}$). The pH (glass electrode Elteca 1220) of feed water inside the chamber, patinated area of the surface (QuickPhoto Industrial, 3.2) and mass gain of the samples were evaluated after the exposure. Samples with patina were documented by the Nikon COOLPIX AW130 camera (Tokyo, Japan). These pictures were cut on sample dimension. Software could recognize pixels with green color of patina and made calculations of the green area. Ion chromatography (DIONEX ICS 1000, eluent: 1.8 $mmol \cdot dm^{-3}$ sodium carbonate +1.7 $mmol \cdot dm^{-3}$ sodium bicarbonate) was used to verify the content of sulphites and sulphates in feed water. Samples of feed water were taken after dosing of $SO_2$ (inlet dose 17.7 $g \cdot m^{-3}$) in steps 5, 10, 15, 30, 45, 60, 90, 120, 180, 240, 300, 360 min.

The second experiment was carried out at 40 °C for 1 week with 8.9 $g \cdot m^{-3}$ $SO_2$ and with different surface pre-treatments. Pre-oxidised samples were also exposed for 24 h with the inlet dose of 17.7 $g \cdot m^{-3}$ $SO_2$. The pre-patinated samples were analysed by X-ray diffraction (XRD, X'Pert Pro, PANalytical, EA Almelo, Holland) and the samples pre-oxidised at 200 °C were evaluated by means of X-ray photoelectron spectroscopy (XPS, ESCAProbe, Omicron Nanotechnology, Taunusstein – Neuhof, Germany) in their original state. The observation of the samples after the exposure was also done by mass gain and patinated area evaluation.

The next experiment was focused on patina formation kinetics and the influence of humidity conditions. Samples were exposed in one week cycle at 40 °C with 8.9 $g \cdot m^{-3}$ $SO_2$. The observed parameters were tested at two levels: continuous condensation and 20 h condensation/4 h drying. Samples were studied by means of spectrophotometry (CM-700d, Minolta, Tokyo, Japan) and scanning electron microscopy (VEGA3 LMU, TESCAN, Brno, Czech Republic) (surface and cross section).

The last experiment was carried out in order to optimise the process. One-week campaigns with cycles of condensation and drying were applied. A different content of sulphur dioxide dose was tested. There were constant contents during the exposure as well as optimal decreasing dosing within the exposure. The contents used in these experiments were in the range of 17.7 to 0 $g \cdot m^{-3}$. Patinas were evaluated by means of XRD and patinated area evaluation. Resistometry was employed for the optimisation of the process, because this method allows a continuous record of the copper track mass loss during the exposure. The method is based on a change in electrical resistance of metal track due to mass loss with time. The resistometric sensor has two parts: reference and measuring. Reference part was covered by epoxy resin. Thickness of the measuring part decreases in time due to corrosion during the exposure and its electrical resistance increase. Adhesion of the patina was tested in a T-bend test, which is a method for evaluating the flexibility and adhesion of a coating on a metallic substrate. Samples with patina were bent around itself in prolonged direction. The experiments are summarised in Table 1.

**Table 1.** Experiments.

| Experiment | Duration | Temperature | SO$_2$ Inlet Dose [$g \cdot m^{-3}$] | Cycle |
|:---:|:---:|:---:|:---:|:---:|
| 1 | 24 h | 40 °C | 8.9–44.3 | Condensation |
| 2 | 1 week | 40 °C | 8.9 | Condensation |
| | 24 h | 40 °C | 17.7 | |
| 3 | 1 week | 40 °C | 8.9 | Condensation Condensation/drying phase |
| 4 | 1 week | 40 °C | 17.7–0.0 | Condensation/drying phase |

Bronze alloys were subjected to patination processes after the optimisation. Two typical bronze alloys were chosen for this experiment and casted: "bell metal" binary alloy 78 wt.% copper and 22 wt.% tin (CuSn22) representing high tin bronze, and "statue bronze" quaternary alloy 90 wt.%

copper, 4 wt.% tin, 4 wt.% zinc and 2 wt.% lead (CuSn4Zn4Pb2) representing cheaper low tin bronzes. Both alloys were cast in a standard electric furnace into a 50 mm thick brass mould at UCT Prague. Chemical composition was checked by X-ray fluorescence (XRF, ARL 9400 XP, Thermo ARL, Chemin de Verney Switzerland) analysis from the core of cast. Samples (50 mm diameter, 5 mm thick) were exposed without any pre-exposed treatment of the surface to compare only influence of composition. Surface was only brushed by brass wire brush.

## 3. Results and Discussion

### 3.1. Influence of $SO_2$ Inlet Dose and pH of Superficial Electrolyte

The experiment began with a study of the influence of sulphur dioxide inlet doses between 8.9–44.3 $g·m^{-3}$ (the level of $SO_2$ in Czech Republic atmosphere is usually in the range 10–100 $μg·m^{-3}$ [9]) for 24 h at 40 °C. Results are summarised in Figure 1. The highest mass gain was recorded in samples exposed to 17.7 and 26.6 $g·m^{-3}$, which corresponded also to patina surface coverages of 68 and 54%, respectively. The patina was spread the most homogeneously at these $SO_2$ doses. The pH measurement of the feed water at the chamber bottom after exposure had the optimal pH range for patination of 3.0–2.7. The pH was 3.2 under the dose of 8.9 $g·m^{-3}$, resulting in a less aggressive electrolyte, lower mass gain and worse patina surface coverage. This low aggressive electrolyte probably did not provide sufficient dissolution of substrate copper for patina formation. The $SO_2$ doses of 35.4 and 44.3 $g·m^{-3}$ provided surface electrolytes with pH lower than 2.4 which led to the fast dissolution of the copper substrate as well as of the patina and mass gain decreased significantly, even to mass loss under the highest $SO_2$ dose. This observation is in agreement with the work of Nassau [18], who proposed the pH value of 2.4 as the limit for patina dissolution.

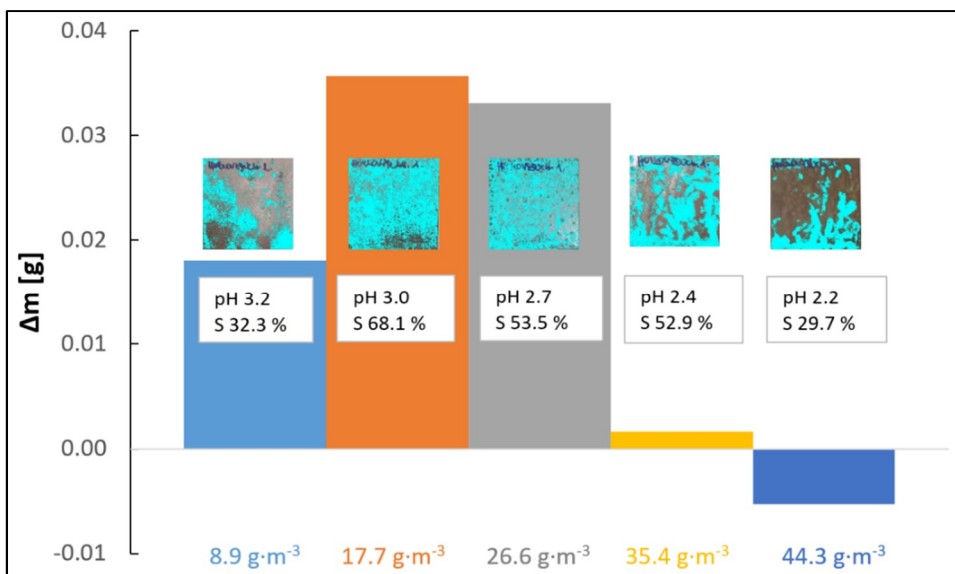

**Figure 1.** Influence of $SO_2$ inlet dose on patina surface coverage (S in %), mass change and pH of surface electrolyte under constant condensation at 40 °C for 24 h.

This first experiment was complemented by two exposures in the chamber without samples with $SO_2$ doses of 8.9 and 17.7 $g·m^{-3}$ for 24 h. Figure 2 presents content of sulphites and suphates within 6 hours of exposure. The pH measurements were recorded continuously during these exposures. The content of $SO_2$ in the gaseous phase was controlled for a dose of 8.9 $g·m^{-3}$ by a data logger (Industrial Scientific, Gas Badge Pro). The decreases of pH values finished within 2–4 h (Figure 3), which probably corresponded to the maximum absorption of sulphur dioxide into water. This observation was confirmed by direct measurement of the $SO_2$ content in the gaseous phase of the chamber (Figure 4).

The values of $SO_2$ content were within 0.6 and 0.9 ppm after 4 h of exposure. Accuracy of data logger according to the manual is 0.1 ppm. This means that the concentration is stable approximately around 2.0 mg·m$^{-3}$ with respect to initial inlet dose 8.9 g·m$^{-3}$.

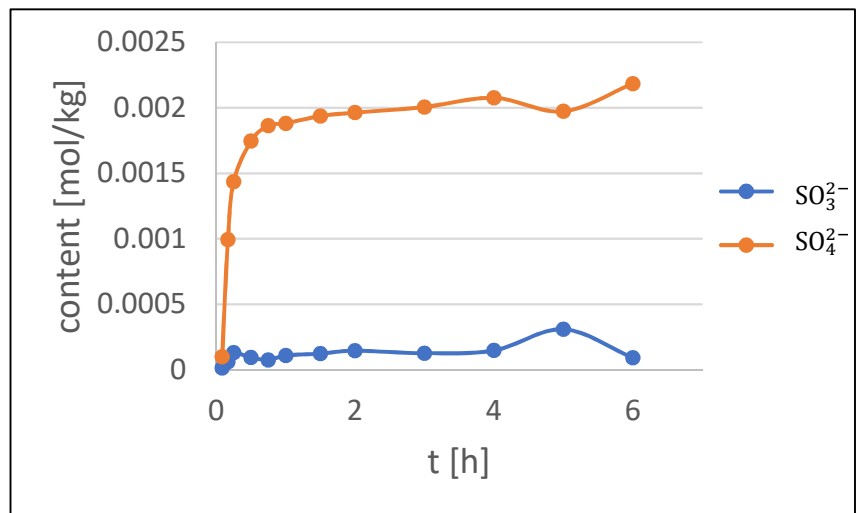

**Figure 2.** Evolution of sulphites and sulphates in feed water.

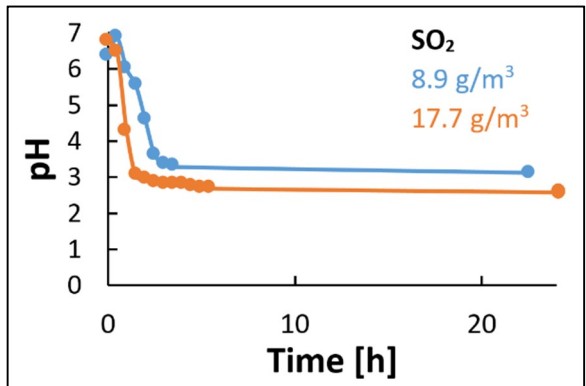

**Figure 3.** Evolution of pH in solution.

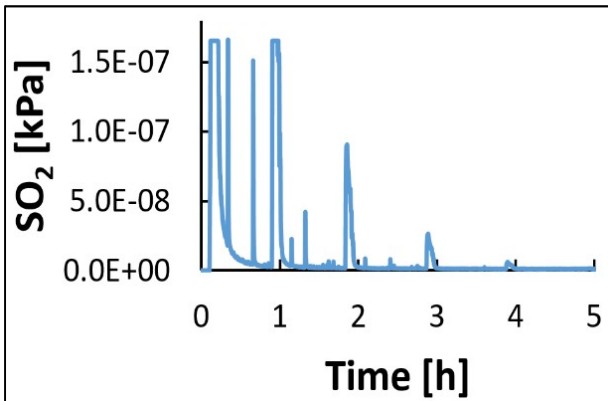

**Figure 4.** Record of the $SO_2$ detection inside the chamber.

In addition, equilibrium thermodynamic calculations were performed within this study to verify the observed experimental data and to assist in explaining them. The calculations were done by means of the PHREEQC (version 3) hydrochemical software [30] with the application of the THERMODDEM (version 1.10) thermodynamic database [31]. In the PHREEQC calculations, the non-ideal behaviour of

the aqueous phase was treated using the Lawrence Livermore National Laboratory aqueous model for activity coefficients [30]. Meanwhile, ideal gas behaviour was assumed for the gas phase, which could be considered reasonable for the system and conditions under investigation. Prior to the equilibrium calculations for the air–water–sulphur dioxide systems of interest, the ability of PHREEQC with THERMODDEM to describe the solubility of $SO_2$ in water at various temperatures and $SO_2$ partial pressures was successfully tested against experimental $SO_2$ solubility data from the literature [32].

There were two data sets calculated for the geometry of the experimental chamber (5 $dm^3$ water and 25 $dm^3$ air). The first data set was calculated for different additions of sulphur dioxide at a constant temperature of 40 °C, whereas the second data set was calculated for a constant $SO_2$ addition of 8.9 $g·m^{-3}$ with temperature increasing from 40 to 90 °C. In addition, both data sets were calculated both with and without allowing possible oxidation of sulphites to sulphates ($S^{IV}/S^{VI}$) in the aqueous phase. The results are summarised in Tables 2 and 3. The pH values measured during the experiments at 40 °C (Figures 1 and 3) corresponded to pH values of the data sets calculated without $S^{IV}/S^{VI}$ oxidation. On the contrary, according to ion chromatography the sulphites content is much lower than sulphates and oxidation occurred (Figure 2). The content of $SO_2$ in gaseous phase for the inlet dose of 8.9 $g·m^{-3}$ decreased significantly, within 4 h, under the theoretically calculated $2.4 \times 10^{-2}$ kPa with the $S^{IV}/S^{VI}$ oxidation suppressed (Figure 4). An oxidation of sulphites to sulphates, therefore, appears to take place in the aqueous phase, given by the presence of oxygen; however, one should take into account the kinetic effects, which may cause redox reactions to run slowly, achieving the calculated equilibrium state after a sufficient equilibration time. In other words, experimental observations being somewhere between the calculations with and without the oxidation may indicate that some processes (e.g., the oxidation) were still running to achieve the equilibrium, although the associated changes were of a small extent, and were thus difficult to be observed/detected experimentally.

**Table 2.** Results of thermodynamic equilibrium calculations for the air–water–sulphur dioxide systems at 40 °C.

| SO$_2$ (g/m$^3$) | Oxidation not Allowed | | Oxidation Allowed | |
|---|---|---|---|---|
| | pH | $P_{SO2}$ (kPa) | pH | $P_{SO2}$ (kPa) |
| 4.4 | 3.4 | $1.9 \times 10^{-3}$ | 3.1 | $6.1 \times 10^{-44}$ |
| 8.9 | 3.2 | $6.9 \times 10^{-3}$ | 2.9 | $4.0 \times 10^{-43}$ |
| 17.7 | 2.9 | $2.4 \times 10^{-2}$ | 2.6 | $2.3 \times 10^{-42}$ |
| 26.6 | 2.7 | $4.7 \times 10^{-2}$ | 2.4 | $6.3 \times 10^{-42}$ |
| 35.4 | 2.6 | $7.5 \times 10^{-2}$ | 2.3 | $1.2 \times 10^{-41}$ |
| 44.3 | 2.6 | $1.1 \times 10^{-1}$ | 2.2 | $2.1 \times 10^{-41}$ |

**Table 3.** Results of thermodynamic equilibrium calculations for the air–water–sulphur dioxide system at various temperatures and a constant $SO_2$ inlet dose of 8.9 $g·m^{-3}$.

| T (°C) | Oxidation not Allowed | | | Oxidation Allowed | |
|---|---|---|---|---|---|
| | pH | $P_{SO2}$ (kPa) | $C_{SO2}$ (mol/kg) | pH | $P_{SO2}$ (kPa) |
| 40 | 3.2 | $6.9 \times 10^{-3}$ | $7.77 \times 10^{-4}$ | 2.9 | $4.0 \times 10^{-43}$ |
| 50 | 3.2 | $1.1 \times 10^{-2}$ | $7.76 \times 10^{-4}$ | 2.9 | $7.2 \times 10^{-43}$ |
| 60 | 3.2 | $1.8 \times 10^{-2}$ | $7.74 \times 10^{-4}$ | 2.9 | $1.3 \times 10^{-42}$ |
| 70 | 3.2 | $2.7 \times 10^{-2}$ | $7.71 \times 10^{-4}$ | 2.9 | $2.1 \times 10^{-42}$ |
| 80 | 3.2 | $4.0 \times 10^{-2}$ | $7.65 \times 10^{-4}$ | 2.9 | $3.4 \times 10^{-42}$ |
| 90 | 3.3 | $5.5 \times 10^{-2}$ | $7.58 \times 10^{-4}$ | 2.9 | $5.2 \times 10^{-42}$ |

Nevertheless, interesting increases in $SO_2$ content in gaseous phase were observed, which corresponds with heater start-ups controlled by the external thermometer above the chamber feed water. The temperature on the surface of the heating rod ($T_{ROD}$) can be calculated according to Equation (8), assuming a heating power equal to heat flux of 500 W, a constant bulk feed water temperature

($T_{WATER}$) of 40 °C, and a heat transfer coefficient for water with natural convection of 500 W·m$^{-2}$·K$^{-1}$. The surface rod temperature of 75 °C is approximately equal to the theoretical overheating of water at the interphase. The overheating itself cannot explain the increase of SO$_2$ in gaseous phase. The calculated values of SO$_2$ content in gaseous phases with oxidation allowed did not exceed very low values (~10$^{-42}$ kPa), even at 90 °C (see Table 3.). Thus, there must be possible reversible SO$_2$ absorption to the liquid phase acidified by the presence of sulphurous and sulphuric acid.

The equilibrium of the system will be different for each chamber geometry and liquid/gaseous phase ratio, etc. However, the process can be controlled by the adjustment of the pH of feed water between 2.7 and 3.

$$T_{ROD} = T_{WATER} + \left(\frac{q}{\alpha \times S}\right) \tag{8}$$

(*q*–heat flux; *α*–heat transfer coefficient; *S*–heating bar surface)

### 3.2. Effect of Copper Surface Pre-Treatment

Three surface states were studied in the first experiment. Samples were exposed in the chamber for 1 week at 40 °C and constant condensation with a SO$_2$ inlet dose of 8.9 g·m$^{-3}$. One set of samples was pre-oxidised at 150 °C for 6 h, the second was pre-patinated in Solution 1 and the third was pre-patinated in Solution 2 (see Experimental part). Pre-patinated samples had significantly thick layers, thus they were analysed by XRD. The layers on samples from Solution 1 consisted mainly of cuprite (Cu$_2$O), while samples from Solution 2 had layers composing chalcocite (Cu$_2$S) (Table 4). The images of the samples before and after the exposure are in Figure 5. The most uniform patina coverage was achieved for the pre-oxidised sample.

**Table 4.** Characteristics of layer obtained by pre-treatment.

| Features | Pre-Oxidation | Solution 1. | Solution 2. |
|---|---|---|---|
| Thickness [nm] | 129 | 737 | 939 |
| XRD–composition | Cuprite Cu$_2$O | Cuprite Cu$_2$O Djurleite Cu31S16 | Chalcocite Cu$_2$S |

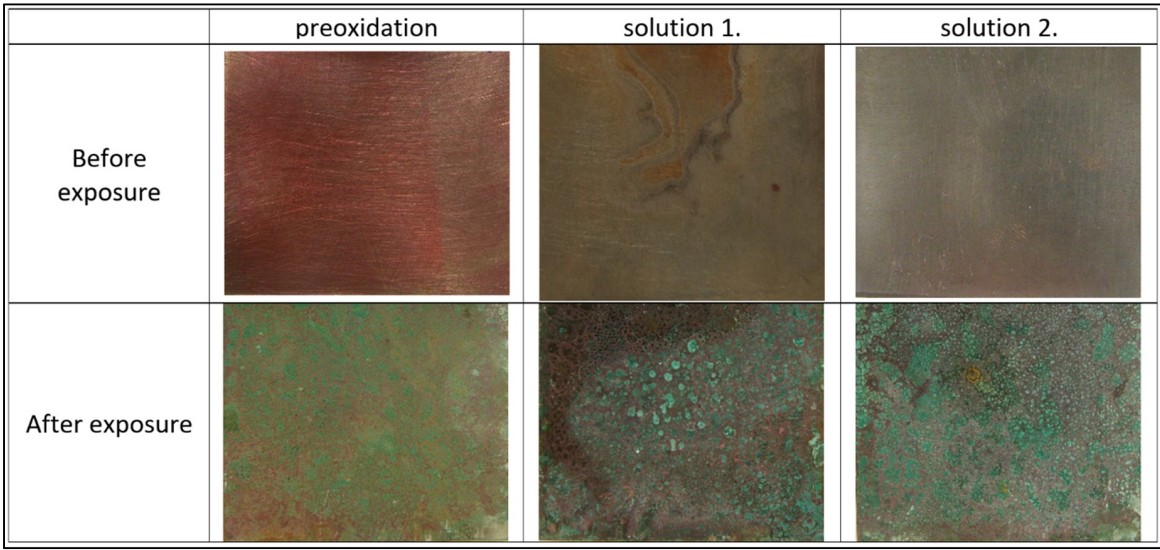

**Figure 5.** Exposure of samples with pre-treated surface.

The next experiments were focused on the pre-oxidation procedure. The temperature was increased to 200 °C and 300 °C in order to reduce the time of pre-oxidation, and different times of pre-oxidation up to 135 min were tested. Samples were exposed for 24 h at 40 °C and constant condensation with a SO$_2$ inlet dose of 17.7 g·m$^{-3}$. The temperature of pre-oxidation of 200 °C allowed

a very sensitive regulation of the pre-oxidised layer. Besides full results at 200 °C, there were results for samples pre-oxidised at 300 °C for 45 min for comparison, as it had the thickest oxidised layer. Results are summarised in Figure 6. The best patina coverage and mass gain was obtained for 200 °C and 25–40 min pre-oxidation. The patina coverage as well as mass gain decreased for longer times of pre-oxidation. The compositions of the oxidised layers were studied by XPS. According to the observed colours, the oxidised layers had thicknesses of 10–350 nm [33–35], thus the Ar sputtering was excluded and only the received surface was analysed. Measured spectra were evaluated in CasaXPS 2.3.15 software (IMFP, RSf etc are part of the sofware library). The data for the chemical state evaluation were obtained from the NIST X-ray Photoelectron Spectroscopy Database. The XPS spectra show three peaks of copper $2p_{3/2}$ on the surface of all pre-oxidised samples before exposure in the chamber (Figure 7). The peak at 932.6 eV corresponds to binding energy of metallic copper and cuprite ($Cu_2O$), the peak at 934.1 eV corresponds to tenorite (CuO) and the peak at 935.5 eV corresponds to cupric chloride ($CuCl_2$) or malachite ($Cu_2CO_3(OH)_2$) [36]. The last peak can be attributed to possible contamination during manipulation. The peak is minor and it did not change with the time of pre-oxidation. The presence of metallic copper can be excluded according to Figure 8 with the Auger spectra of copper $L_3M_{4.5}M_{4.5}$ peak. The kinetic energy peak at 916.9 eV corresponds to cuprite and the peak at 917.6 eV corresponds to tenorite. The peak of metallic copper would be situated at 918.6 eV [36,37], and it is missing, thus the layer is continuous.

There was obviously in the XPS spectra (Figure 7) an increasing ratio between tenorite and cuprite with time of oxidation (up to 75 min) and consequently the ratio is constant. Tenorite is in general less soluble than cuprite, thus it has an inhibitive effect on oxidised layer dissolution and the patina formation. The cuprite layer at 10 min pre-oxidation was too thin, thus the optimal time for pre-oxidation was 25–40 min.

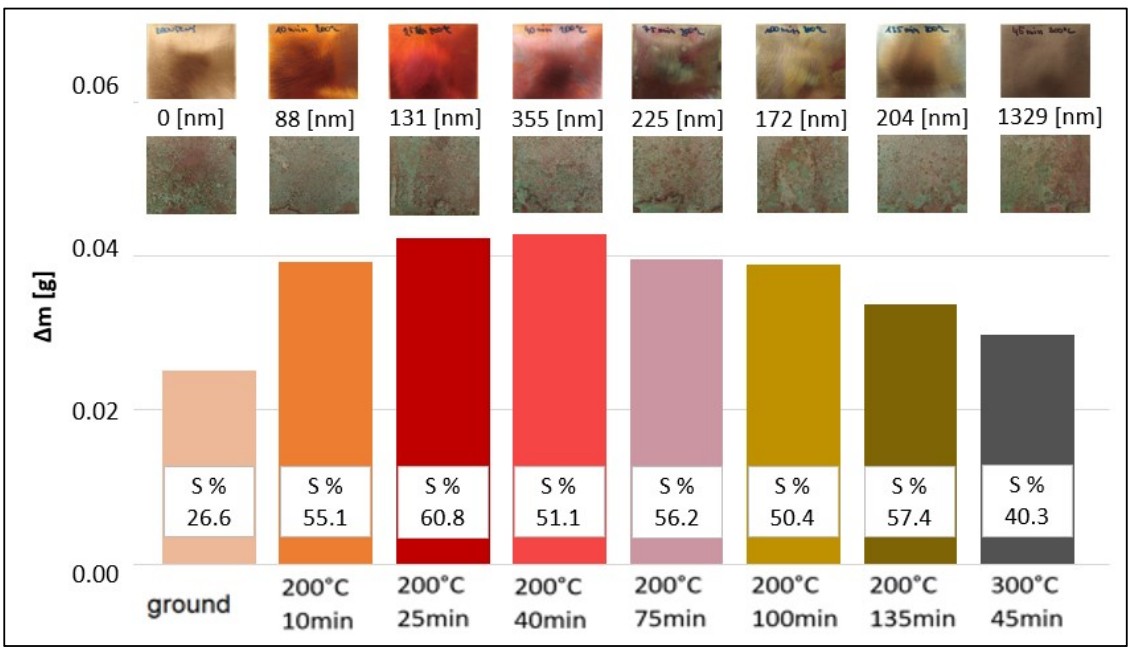

**Figure 6.** Mass gain and surface coverage (S %) on samples pre-oxidised at 200 and 300 °C for different times [thickness of oxidic layer] after 24 h of exposure at 40 °C with a $SO_2$ inlet dose of 17.7 g·m$^{-3}$.

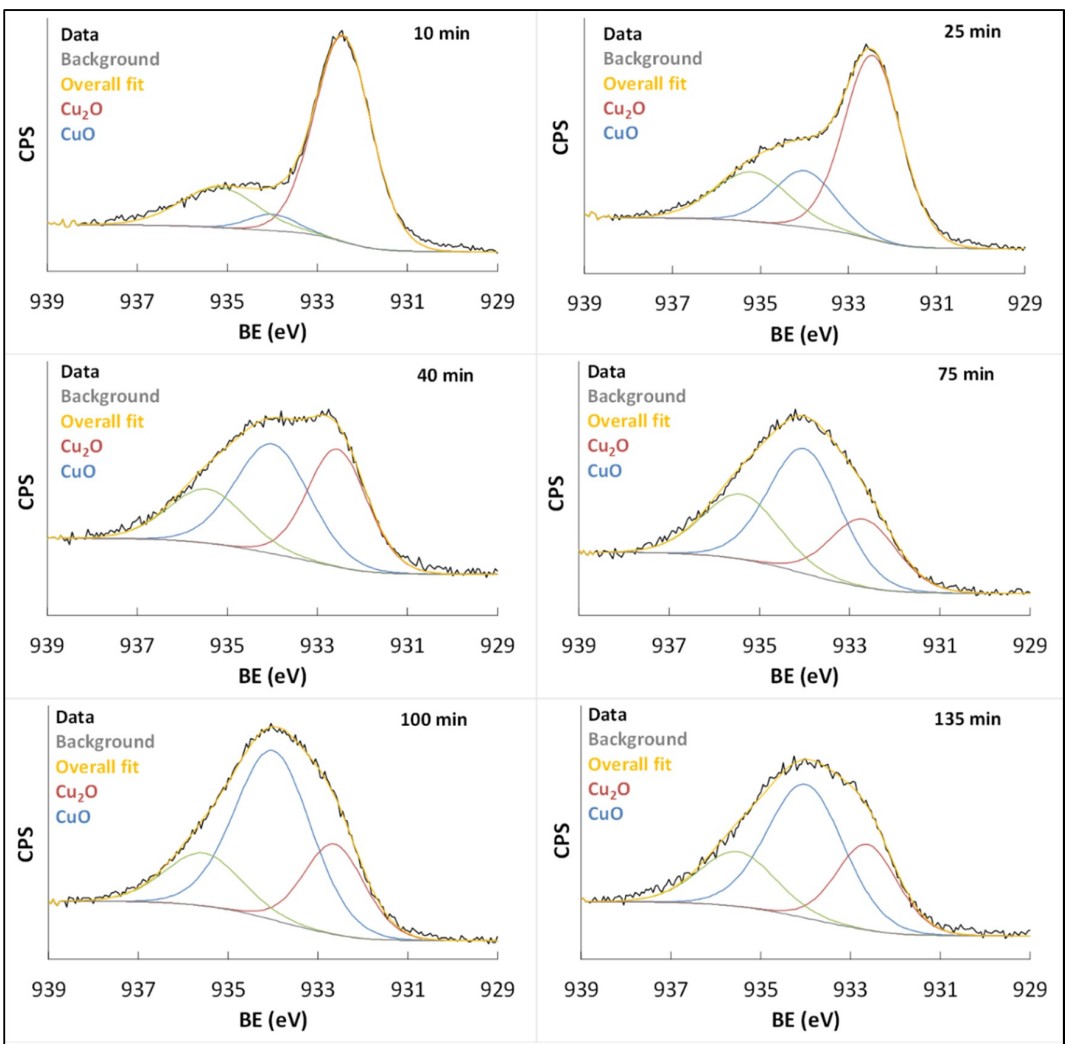

**Figure 7.** X-ray photoelectron spectroscopy (XPS) peaks of copper $2p_{3/2}$ (binding energy) of samples pre-oxidised at 200 °C for different times (10–135 min).

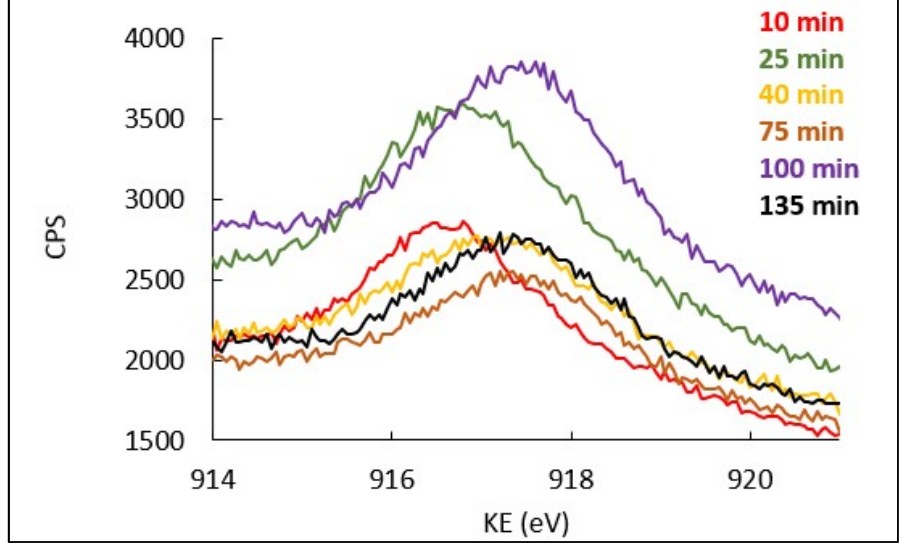

**Figure 8.** Auger peaks of copper $L_3M_{4.5}M_{4.5}$ (kinetic energy) of samples pre-oxidised at 200 °C for different durations (10–135 min).

### 3.3. Kinetics of Patina Formation and Influence of Drying Phase

One-week exposures at 40 °C with a $SO_2$ inlet dose of 8.9 g·m$^{-3}$ were performed in this part of the work. The first experiment was carried out with constant condensation. The second consisted of four cycles (over four days) 20 h condensation at 40 °C with $SO_2$ doses and consequently 4 h of drying in ambient atmosphere. These four cycles were followed by 3 days of constant condensation. The loss of metallic copper during these experiments was recorded by resistometry (Figure 9). The record showed continuous but decelerating dissolution of substrate copper during continuous condensation exposure. The process with drying stage showed as fast dissolution as continuous condensation but stopped during the drying stage, thus allowing the formation of patina nuclei at different sites promoting better homogeneity of lateral coverage.

The results of all these exposures are summarised in Figure 10, as sample colours were evaluated by spectrophotometry. Samples were compared to the patina of the roof sheets from Queen Anne´s Summer Palace in Prague, which is more than 300 years old. Patina formed under constant condensation conditions, provided less uniform patina coverage compared to a cycle when a drying stage was involved. With cyclic condition the patina coverage was better, and the colour approached a natural patina.

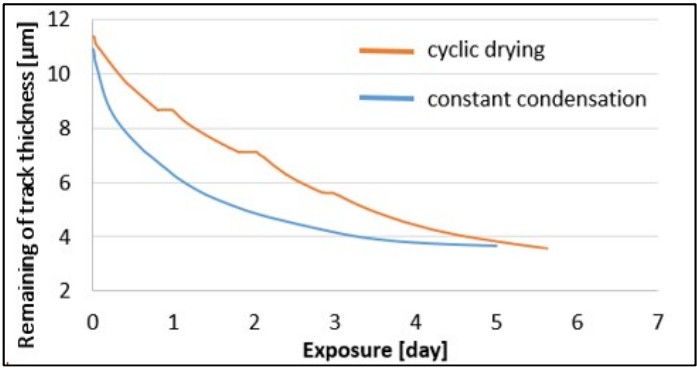

**Figure 9.** Resistometric record during exposures.

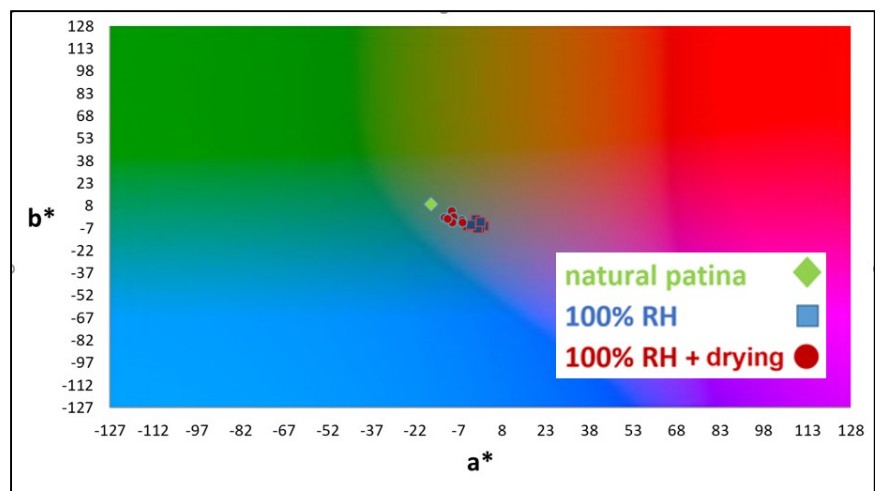

**Figure 10.** Spectrophotometry results displayed in CIELab coordinates and their comparison to natural patina, a—the red/green coordinate, b—the yellow/blue coordinate.

Samples obtained from exposure with constant condensation and cycling with drying stages were evaluated in detail by scanning electron microscopy (SEM)/energy-dispersive X-ray spectroscopy (EDS). Samples of the exposure with drying stages were collected after each drying stage. The experiment with constant condensation was complement with two separate exposures for 2 and 4 days (Experiment 3).

Appearance, patina morphology and cross-section are compared in Figures 11 and 12. A thin layer of cuprite (approx. 1 μm) was formed under constant condensation within 2 days of exposure (Figure 12). Sulphate based patina started to form after 4 days exposure above the former cuprite layer. These compounds were identified by XRD as the brochantite precursors strandbergite ($Cu_5(SO_4)_2(OH)_6 \cdot 4H_2O$), also called kobyashevite, and posnjakite ($Cu_4(SO_4)(OH)_6 \cdot H_2O$). They have flower-like morphology, which is similar to the natural patina formation described by FitzGerald [8]. These flower crystals grow in the latter exposure and they form locally thick islands of patina up to 30 μm after 1 week of exposure, when brochantite ($Cu_4(SO_4)(OH)_6$) begins also to be present in the patina. The kinetics of patina formation were much faster with drying stages inserted to the patination process. It allowed the formation of new nucleation sites at the surface after each new inlet dose of $SO_2$. The increasing number of nucleation sites allowed faster and laterally more homogeneous growth. There were obvious islands of sulphates already after 2 days' exposure. The number of islands was much higher when compared to previous constant condensation after 4 days. The islands had thicknesses of approximately 20 μm, and since the surface was covered significantly by them, the consequential growth was rather lateral. The thickness did not grow further and patina uniform in thickness was formed.

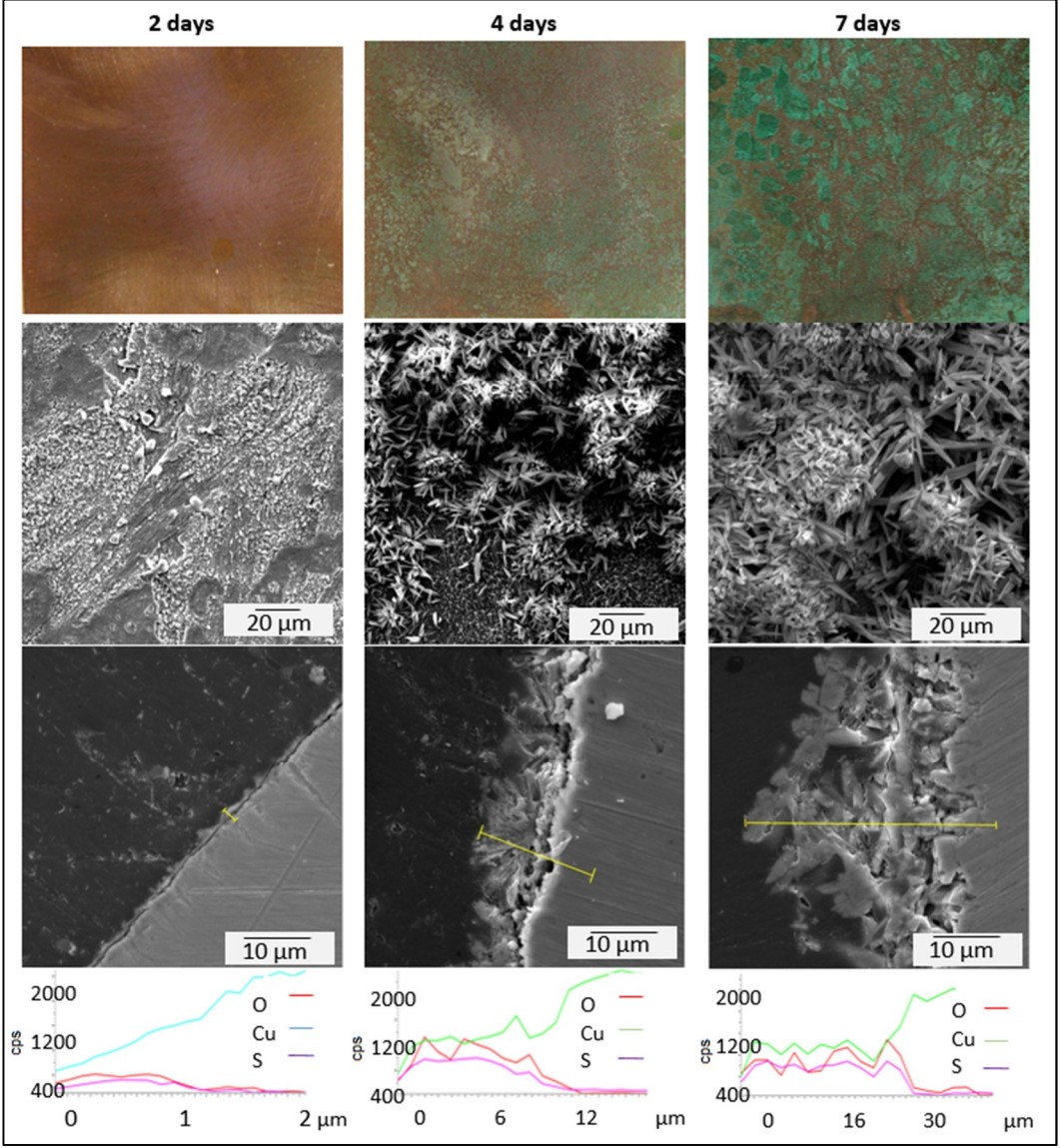

**Figure 11.** Surface appearance, patina morphology and element concentration profile of samples exposed for 2, 4 and 7 days in process with constant condensation (without drying stages).

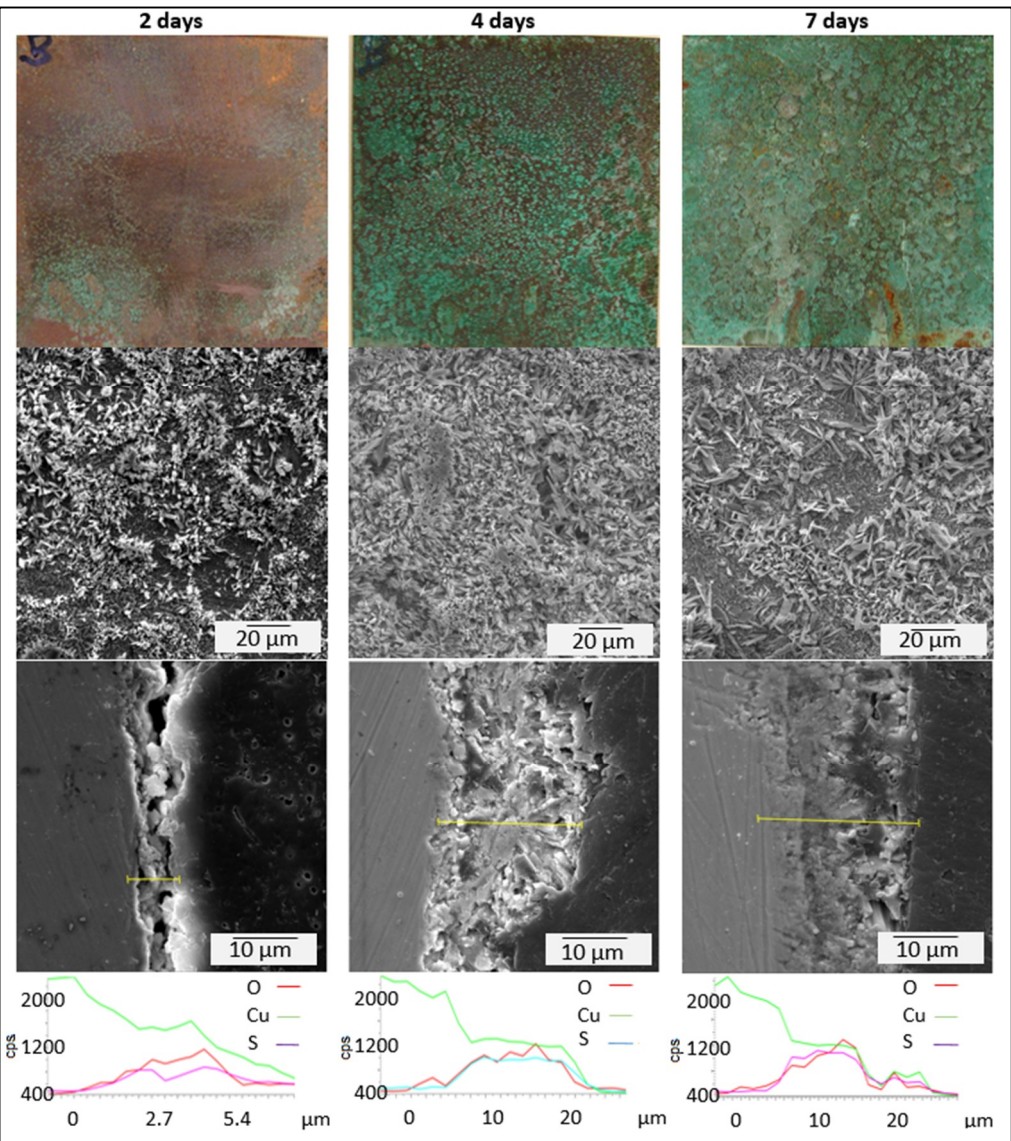

**Figure 12.** Surface appearance, patina morphology and element concentration profile of samples exposed for 2, 4 and 7 days in process with drying stages.

### 3.4. Optimisation of Patination Process

Decreases of pH values below 2.7 were recorded during cycling experiments when a new inlet dose of $SO_2$ 17.7 g·m$^{-3}$ was injected after each drying stage. The new optimised process was verified on samples pre-oxidised for 40 min at 200 °C. It began with a standard inlet dose of 17.7 g·m$^{-3}$ and continued with decreasing inlet doses of 8.9, 4.4 and 0.0 g·m$^{-3}$ after each drying stage. However, before a final 3-day cycle without a drying stage, a higher inlet dose of $SO_2$ 17.7 g·m$^{-3}$ was introduced again. The process is described in Figure 13, which also contains the record of a resistometric copper probe. Resistometric probes have been described in detail elsewhere [38–50]. The metallic trace corroded approximately 1 μm in the first cycle, but all consequential cycles dissolved only 0.3 μm of metallic trace. The pH was kept in the range of 2.7–3.0 during the entire experiment. This optimised process was verified for reproducibility in four independent replicates. The surface appearance can be seen in Figure 14. The patina was well attached to the copper surface. The T-bend test was used to test the adhesion of patina and no cracks were observed after the third bend (Figure 15). The patinated area was approximately 80%. The surface coverage was uniform, and it contained mostly brochantite (see Figure 16).

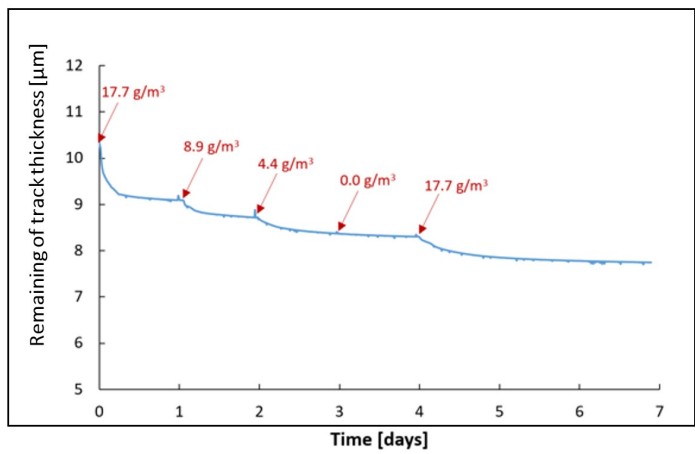

**Figure 13.** Resistometry record on copper probe during optimised process.

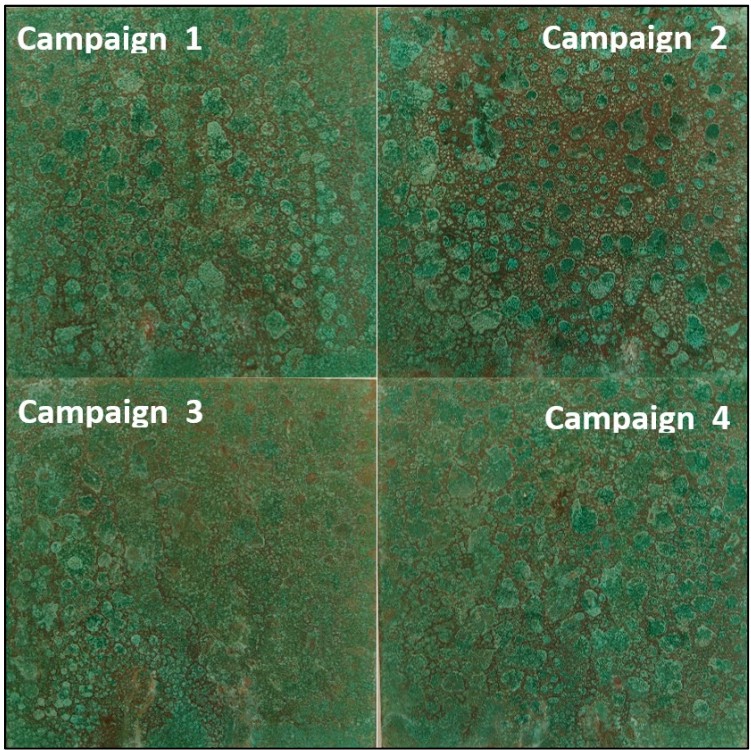

**Figure 14.** Examples of sample (5 cm × 5 cm) appearances after exposure in the optimised process in four different replicates.

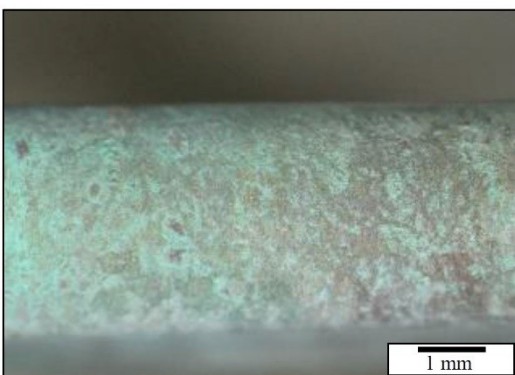

**Figure 15.** T-bend test, 3rd bend.

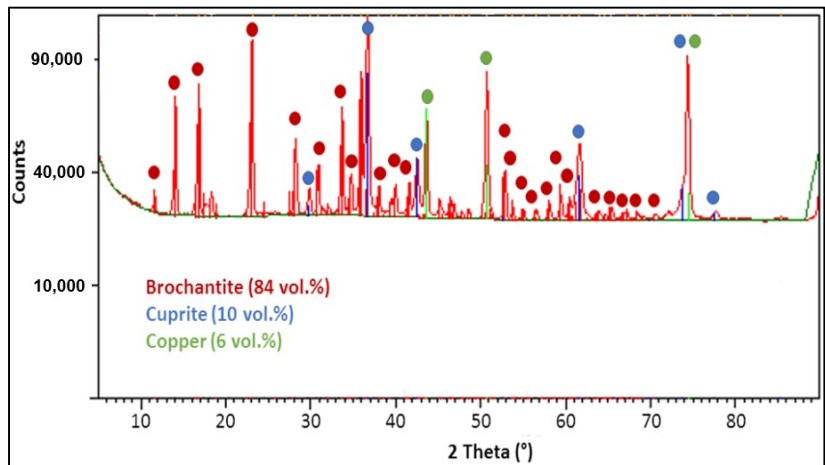

**Figure 16.** Example of diffraction patterns after optimised process patination.

Examples of bronzes patinated in an optimised process are given in Figure 17. Patina formation is significantly slower on a bronze surface, but patination is possible and the prolongation of the process to several weeks could provide better coverage.

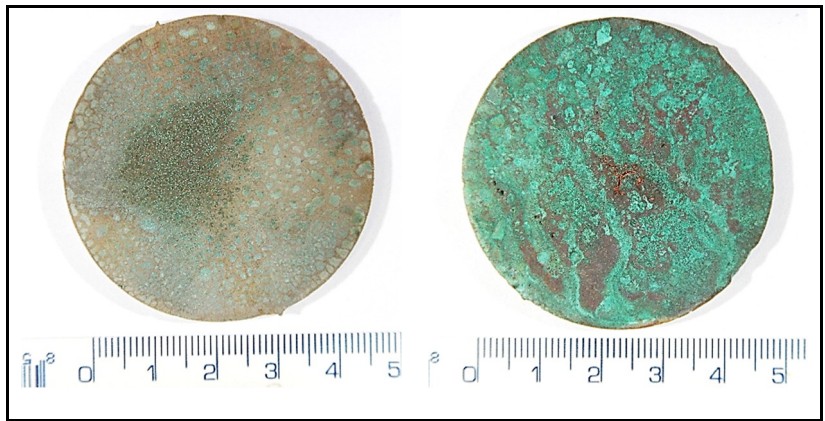

**Figure 17.** Artificial patina on the surface of experimental bronze alloys samples (ø 50 mm), Experiment 4: "bell metal" CuSn22 (left) and cheap version of "statue bronze" CuSn4Zn4Pb2 (right).

## 4. Conclusions

It was possible to obtain a well-adhering artificial patina based on brochantite prepared by the proposed process within one or two weeks. That was a much shorter time compared to several years in natural atmospheric conditions. The ideal process includes a condensation stage at 40 °C for 20 h and drying stages in ambient atmosphere for 4 h. The optimal pH of the feed water should be kept within 2.7 and 3, which is provided by the sulphur dioxide dosing decreasing from 17.7 g·m$^{-3}$. Patina lateral homogeneity was supported significantly by surface pre-oxidation at 200 °C for 40 min.

**Author Contributions:** R.B. and J.S. conceived of and designed the experiments; R.B. and J.S. are responsible for writing—review and editing, P.R. provided methodology and materials, M.K. provided thermodynamic equilibrium calculations, J.F. and K.J. provided formal analyses and analysed the data.

**Funding:** This research was funded by Czech Ministry of Culture in programme NAKI II, grant number DG16P02H051.

**Acknowledgments:** The authors are very grateful for the support from Czech Ministry of Culture in programme NAKI II under the project number DG16P02H051.

**Conflicts of Interest:** The authors declare no conflict of interest.

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
