# Peer review of "Artificial Patination of Copper and Copper Alloys in Wet Atmosphere with Increased Content of SO2"

_coatings, doi:10.3390/coatings9120837_

Round 1

Reviewer 1 Report

Review Artificial patination of copper from gaseous phase

The article presents a study on brochantite formation from gaseous phase on bronze substrates. The authors used different methods to monitor patina growth and structure changes. The methods that were used were optical and Electron microscopy, XPS and XRD as well as resistometric method to follow the concentration of dissolved copper.

The paper contains 12 Figures, 2 tables and 42 references.

The paper is of important topic in scientific and cultural heritage field.

The comments are given bellow.

Title: Authors conducted a study on bronze samples, but title is Artificial patination from gaseous phase. Should you change it to bronze or at least copper alloys?

Abstract:

 Line 20: it was possible to obtain brochantite- You can rephrase to brochanitite was deliberately growm

Line 21: say approx. thickness

Introduction:

Line38: these products(patina) are insoluble products

Line 53: in the next step, natural patination, insert natural if that is the case.

Line 76: Insert a reference on brochantite formation in sheltered areas

Line 79: say that this brochantite formation is artificial process..

In introduction in the end paragraph, the gap sentence (what is new compared to previous data in literature) what are the goals and methods of this study.

Materials and Methods

Lines 128-148

Description of the experiments could be presented in table  for more clear distinction between the experiments

Line 153: Where were bronze samples casted, how was chemical analysis done and checked?

Results and discussion

Avoid saying in the next experiment (throughout the text)

Line 174- previous experiment, which one?

It is difficult to follow due to such description /structure of experimental description.

Say: experiment with constant condensation or experiment with drying cycles like in Figure 8.

Lines 182-222 were probably written by different author. Please make a clearer/shorter part as it is now, more consize. Also, bring Table 1 and2 closer to this text. Now, when it is on the other page, it is difficult to study this part of the paper.

Line 239: Which preparenated thick layers( what is the structure, what type of patina was achieved). The information s already needed at this part of sentence.

Line 250: What is the structure of ths patina or refer to figure or information given later on in the text

Lines 253-258 where XPS data is given, please explain why peak for Cu2O at 935.5 in your figures/analysis are not at 935.5 eV but at933 eV.

Line 271

Is figure 5 ok, why are the coloumns/surfaces squizzed?Looking from the pen description at the top of the samples?

How was XPS data fitted?

Line 280: Figure is not graphically nicely presented (too big fonts in comparison to other figures), also the colours are  difficult to differentiate, maybe write the time directly on the curve?

Line 306: remainings of track change step

 Line 308:

This figure could be smaller, fonts bigger and not extended to both edges of the paper

Figure 9: Explain a nd b axis a bit..

Line 318: from which experiment?

Line 327: faster with drying stages during is better expression

Figure10: Figure captions should be at the bottom of figure. Explain the lower graphs in Figure 10.Legend is strangely hidden, not clear.

Lines 344: which previous experiment?

Line 365: Figure 13- insert a scale or define magnification in figure caption

Figure 14. not a nice resolution of a figure 14.

Figure 15

Bronze photos are not clear, they are in Frame, why? Artificial brochantite patina? In which experiment?

Figure 14. not a nice resolution of a figure 14. Figure caption should be bellow Figure on the same page.

Figure 15

Bronze photos are not clear, theyare in Frame, why? Artificial brochantite patina? In which experiment?

Conclusions

Not the best conclusions- explain which type of experiment gave best brochantite patina (in terms of coverage, homoeginity, appropriate thickness and time) meaning explain which experiment (Drying, constant humidity???

Authors contribution: Use past tense: provided

References: Some references have all authors written some have et.al. Unify. Also, unify giving page numbers p x-y.

Author Response

All replyes are attached in pdf document. English language editing was done also as you recomended. Thank you for your comments.

Richard Bureš

Reviewer 2 Report

The article deals with a systematic study of patina formation in polluted environment. The study has been conducted in a logical way. There are only some issues that need to be additionally described or improved:

line 68 – compared  instead of compere

lines 111-120- more detailed description of condensation chamber, how were the samples positioned in the chamber? It is not clear if the first set of experiments was conducted under continuous condensation or under some other conditions?

- please give some references about resistometry technique or describe it in more details. Is this electrical resistance method or some other method?

-please give some references about CIELab coordinates

Author Response

(The authors gave the same response as above.)

Reviewer 3 Report

The paper investigates the possibility of artificially patinating copper and bronze by inducing the formation of a stable patina based on brochantite through artificial ageing in a humid atmosphere containing SO2.

This topic deserves to be studied because methods to obtain reproducible patinas in terms of chemical and physical features are currently few and properly standardized methods are lacking.

The paper is organized according to a classical sequence of sections, however not all of them seem to contain all the needed information or to be well-developed. Different aspects were considered and tested, but not all of them are deeply investigated. Some methodological choices are not immediately clear and a systematic and organic approach seems to lack in the experimental part as well as in the discussion of the results. Sometime information concerning the methodology followed is only reported in the results and discussion section.

Main issues are reported in the following.

Title

The title points the attention on patination from gaseous phase. However, from the experimental part it seems that patination occurs not in presence of a “simple” gaseous phase but in presence of condensed water containing dissolved SO2 and possibly sulphate. As a consequence, in my opinion, the title could be misleading.

Introduction

Lines 85-89: the reference [6] is about archaeological bronzes and does not seem to contain information related to SO2 concentrations limiting brochantite formation. Moreover, being the paper about the formation of a brochantite-based patina, the basics provided about brochantite formation/stability (SO2 concentration limits, ranges of stability, …) could be more specific and detailed.

Lines 94-95: “Homogeneous oxidation….radical.”. Please revise this sentence and its meaning because, in the present form, it is not scientifically correct and does not respect the content of the original source ([25]). In fact, if the oxidation is homogeneous, it does not take place in an aqueous phase, but it occurs in the gaseous phase due mainly to the action of hydroxyl radical. This leads to the formation of gaseous SO3 that subsequently dissolves in the aqueous phase.

Lines 96-100: in the heterogeneous oxidation, the solubility of SO2 is also increased by the oxidation process itself. 

At the end of this section, the aim of the work should be better pointed out. In particular, the specific reasons for the following aspects should be clearly explained: (a) why the study was carried out (e.g. in which fields and why artificial brocantite-based patinas are needed) and (b) why patination from gas phase was considered with respect to other possibilities.

Materials and Methods

Line 112: “specific conditions were required” this sentence seems to be too generic for this section. The specific conditions required should be mentioned.

Lines 113-115. More details on the chamber should be provided: is the chamber completely home-made or is it a commercial chamber implemented by the authors? Which material the chamber is made of? If the chamber is home-made, how are temperature and humidity controlled? How are the blowing air and the SO2 volume dosed? Can be and is SO2 concentration kept constant in the atmosphere of the chamber during the test or a dose is supplied at the beginning and then the system is left to evolve? Which was the volume of feed water?

Line 116. At which temperature was the saturated solution of sodium sulphate maintained to reach 98% of RH? Can some reference be provided about the usage of this specific salt to control RH and the RH levels that can be reached as a function of temperature?

Line 119. How copper sheets were placed inside the chamber?

Lines 124-127. On which basis and to obtain which kind of corrosion layer were the pre-patination solutions selected? Were they already used for copper pre-patination?

Lines 128-131. See comment on lines 113-115 about SO2 concentrations (the same comment should be considered for all the tests described). If humidity in the chamber is regulated only by the presence of a water vessel at the bottom of the chamber (as it seem from the description at lines 115-118), RH during this test will be always 100%; this condition of continuous condensation should be specified at this point.  How was the patinated area observed and the patina surface coverage (%) calculated? The software used to analyse the images is reported but not the microscope used for tacking the images. Were compositional analyses performed to ensure that only brochantite forms and no other or a mix of sulphate compounds?

Lines 132-136. Was, in this test, final pH controlled? Were compositional analyses performed after the ageing? Also in this case the condition of continuous condensation should be specified.

Lines 137-141. The choice of a test level characterized by continuous 98% relative humidity should be better justified. This RH level is very close to the continuous 100% RH level of the first step: were tests for verifying the stability of this RH level performed? How it was controlled that a no continuous condensation took place also in this case? What about SO2 dissolution and pH when a saturated sodium sulphate solution is used instead of demineralized water (see also the subsequent comment about Lines 290-291)? Were compositional analyses performed (i.e. EDS, XRD…)?

Lines 142-148. A random approach, as described in the text, does not seem appropriate for optimising a process. How the content of SO2 was randomized? Was this content determined? Why was it varied between 17.7 and 0 mg/m3 if in the results a dose lower than 8.9 mg/m3 was judged too low for forming an homogeneous patina? Was, in this test, pH controlled? Further details about resistometry measurements should be given at this point instead of at the end of the paper (§3.4).

Lines 149-153. How many samples and of which size were patinated? To which aim and which kind of analyses were performed after the patination?

Results and Discussion

3.1

Considering that in this test no specific compositional analyses are reported, an experimental demonstration of the brochantite formation seems to lack, as well as a support to the following sentence reported in the abstract: “The higher sulphur dioxide content above 17.7  g.m-3 was too aggressive a surface electrolyte, which led to the dissolution of the brochantite”.

Lines 174-end of the §3.1: this part seems more related to the set-up of the chamber and to the description of how the ageing conditions to which the specimens are exposed evolve. It should anticipate the other results and eventually used to justify or discuss them.

Line 180. Which was the detection limit?

Line 203-208: the oxidation rate of sulphites to sulphates could be experimentally verified by analysing the feed water by Ion Chromatography.

Fig.3: how many SO2 remains in the atmosphere of the chamber after 4h with respect to its initial content and with respect to a typical urban concentration?

3.2.

line 239-243: how thick were the layers? It is not clear if the mentioned layers (and their composition) are those obtained after pre-patination or after pre-patination + ageing with SO2. In any case the characterization of the specimens in one of the two conditions (i.e. before or after ageing) is missing. What about composition of the “patina achieved for the pre-oxidised sample” and, also in this case, is “patina” the layer obtained after pre-oxidation or after pre-oxidation and ageing?

Lines 244-246. Which combinations of temperatures and times were exactly tested? Line 249: how thick was the thickest oxidised layer?

Fig.5: this figure seems quite confusing. It’s not clear the reason of using stretched images of specimens as mass gain bars. If these images represent the specimens before the test, they should be inserted as images with the right proportions. In the caption the symbol used for indicating the surface coverage should be repeated.

Even in this part it is not clear if all the investigations refer to the pre-oxidised specimens or to the pre-oxidised specimens after ageing with SO2. Fig.6 the green line is not defined.

A conclusion for §3.2 should be reported. How the pre-treatments tested influence the brochantite-based patina formation?

3.3.

Considering the experiment description, it seems that in the condensation/dry test a new dose of 17.7 mg/m3 is supplied after each dry phase while in the other two tests only one dose is supplied at the beginning; if this is true, the total SO2 amount which the specimens are exposed to are different in the three tests that want to be compared. How can this difference affect the results?

Lines 290-291 see the previous comment about lines 137-141. Moreover, how the dissolution of SO2 and pH changes in a saturated solution of sodium sulphate with respect to water? This condition was not considered in §3.1 when equilibrium calculation are discussed. How can it influence the results?

It could be interesting to show SEM or MO images also for specimens exposed at RH 98%. Line 317: “These were necessary to …” this sentence is not clear. Appearance, morphology and cross sections observations were not mentioned in Materials and Methods section.

3.4.

Line 351: “…dissolved only 0.3 um.” of what?

Lines 352-356. How was the adherence of the patina tested? Some details about variability of the quantitative analyses performed on the four independent replicas should be provided.

Lines 357-359. Results and discussion about bronze patination is really poor, it consists in three lines and Fig.15 showing two images of the surfaces. About Fig.15, please add some detail about images (e.g. technique, magnification,…?)

Minor comments:

Please revise commas (es. Line 47 “by, direct oxidation”); Line 62 different typefaces; Line 68 “compere”; Line 112: “studies of the”.

Author Response

(The authors gave the same response as above.)

Round 2

Reviewer 3 Report

Responses to reviewers were prepared with poor care, also with regards to the English language.

Responses to reviewers were prepared with poor care, also with regards to the English language.

The text was somewhere ameliorated, but some main issues and discussion remain not completely improved. Moreover, several requested details are only mentioned in the response to reviewers, without any comment in the text. Major issues are listed below:

1) Lines 120-122: “These properties are given by the composition of patina and the artificial patina form in condition of atmosphere with increased SO2 content would be  based on sulphates like the natural patina form usually in industrial areas.” Form or formed? Please revise the English

2)Line 128. The material of the chamber is still not reported.

3)Lines 140-145. Authors reply to the reviewers comment about this part (“Lines 124-127. On which basis and to obtain which kind of corrosion layer were the pre-patination solutions selected? Were they already used for copper pre-patination?) that they “ Correct in the text.”, but the text here is exactly the same as in the previous version. They only reported some info in the reply to reviewers. Then, I noticed they added something in results and discussion, but, in my opinion, the reason of the choice should be explained in materials and methods.

4) Line 146 and table 1. The duration of the first test is changed with respect to the original version (it is reported that it “started with 24h” instead of “started with 1week”). This change is not justified in the response to reviewers nor highlighted in red in the new version.

Moreover in results and discussion, at lines 183-185 it is stated: “The experiment began with a study of the influence of sulphur dioxide inlet doses between 8.9–44.3 g·m-3 …. for 1 week at 40 °C.” In fig.1 they mention again a duration of 24h, but the results in the figure are exactly the same than in the previous version of Fig. 1 when a test of 1 week was indicated…. what exactly was done??

5) The same at line 159 for the SO2 concentration of the last experiment (8.9 instead of 17.7 g/m3).

6) Line 170: details of the T-bend test?

7) Authors state to have “Accepted. Correct in the text” the comment about details on resistometric measurements, but the text seems to be unchanged as regards this aspect.

8) Lines 175… Details on prepatination replicas were not added  in the text

9) Line 208. Detection limit was not added in the text

10) Line 227. Did the authors really used Gas Cromatography for determining sulphites and sulphates?? What were the experimental conditions?? Moreover, authors now state: “ The pH values measured during the experiments at 40 °C (Figure 1. and 3.) corresponded to pH values of the data sets calculated without  SIV/SVI oxidation. Gas chromatography was used to verify the content of sulphites and sulphates in  feed water (Figure 2.).” a) No comments are reported about this new figure. b) According to chromatographic results, it seems that oxidation occurred, as sulphides result much lower than sulphates.

11) Information requested about Fig.3 (previously fig.2) was not inserted in the text

12) Line 269. “Samples were exposed in the chamber for 1 week at 40 °C and constant condensation with a SO2 inlet dose of 17.7 g·m-3”. This does not correspond to the new Table 1 and the new description of the tests, where a concentration of 8.9 g·m-3 seems to have been used for this 1 week test

13) Line 281. “Samples were exposed for 24 hours at 40 °C and constant 281 condensation with a SO2 inlet dose of 17.7 g·m-3”. In the previous version this test was conducted for 1 week, but the results are the same in both the versions. Why the author did not mentioned the reason of this change in the reply to reviewers?

14) Fig.7. green line label was not added

15) §3.2. How was XPS data fitted is again reported only in response to reviewers and not in the text.

16) Line 323. “One-week exposures at 40 °C with a SO2 inlet dose of 8.9 g·m-3 were performed in this part of 323 the work.” In the previous version of the text the inlet dose reported was 17.7 g·m-3. This change is again not highlighted in red and results are the same as in the previous version.

Author Response

Thank you for comments. I hope that all replies will be clear and corect.

Round 3

Reviewer 3 Report

Line 25: forms a too aggressive…